# DUAL-COMBINER NETWORK WITH MULTI-ATTENTION FOR COMPOSED IMAGE RETRIEVAL

## ABSTRACT

Composed Image Retrieval (CIR) is a challenging task that aims to retrieve target images based on multimodal queries consisting of a reference image and modifying text. Due to the semantic and modal gaps between images and text, existing CIR methods struggle to accurately compose reference images and modifying text. Although some of these methods can establish fine-grained correspondences between local text tokens and visual regions, they often focus on text-specified content in the reference image, which overlooks the consistency of unmentioned regions with the target image. To address these limitations, we propose a novel Dual-Combiner with Multi-attention (DCMA) network that employs a progressive fusion strategy to integrate self-attention, cross-attention, and channel-attention mechanisms for well capturing the user's query intent. Specifically, the global combiner leverages the multi-attention framework to capture global context information of the users' query intent. In parallel, the local combiner focuses on fine-grained information to accurately localize modifier areas while preserving the consistency of other regions between the reference and target images. Therefore, the proposed DCMA can precisely encode multi-granularity multi-modal query information into the fused representations by the multi-attention framework. Extensive experiments demonstrate that DCMA achieves state-of-the-art results across multiple benchmark datasets, validating its effectiveness in capturing complex multi-modal interactions for composed image retrieval. The source code for this work will be available later.

## 1 INTRODUCTION

Traditional image retrieval typically employs two standard approaches: Image-based retrieval and Text-based retrieval Li et al. (2025); Fei et al. (2025); Duan et al. (2025); Pu et al. (2025). The former retrieves visually similar images using an input query image, while the latter matches semantically relevant images based on textual descriptions. However, with the exponential growth of multimodal data on the internet, image data now exhibits inherent characteristics of massive-scale similar images and diverse sourcesWang et al. (2025b). Concurrently, user demands for both retrieval precision and query flexibility have significantly increased. Against this backdrop, the limitations of unimodal retrieval methods have become increasingly prominent: image-based retrieval struggles to capture the abstract semantics of images or users' subjective intentions, while text-based retrieval is constrained by the accuracy and comprehensiveness of textual descriptions, making it difficult to convey rich visual details within images. To address these challenges, composed image retrieval, which allows users to fuse an image and a short text as the query for more accurately expressing complex search intents, has emerged as a novel paradigm Du et al. (2025); Song et al. (2025).

Recent research in CIR has explored diverse multimodal fusion strategies to enhance query representation Wen et al. (2024); Bai et al. (2023); Wen et al. (2023); Tang et al. (2024); Karthik et al. (2023). Despite these advances, CIR remains highly challenging due to the inherent semantic and modal gaps between images and text, which hinder accurate modeling of joint retrieval intent. As illustrated in Figure 1, existing approaches often struggle to execute the global semantic transformations specified by the modifying text (e.g., "yellow and grey stripes"). However, they frequently fail to preserve the fine-grained details from the reference image (e.g., clothing style). Meanwhile, the excellent performance of Vision-Language Pre-trained Models(VLPMs) such as CLIP and BLIP Radford et al. (2021); Li et al. (2022)has made them a promising direction for CIR. However, the

effectiveness of such strategies remains limited by the scale and quality of existing CIR datasets, while the substantial computational cost of fine tuning VLPMs further restricts their practicality.

To address the limitations of existing methods, a Dual-Combiner with Multi-attention (DCMA) network is designed. We observe that the output of CLIP's intermediate layers can provide rich local information for detailed region-text matching, while SigLIP demonstrates strong ability to capture discriminative global contextual semantics. In this paper, we first synergistically integrate the intermediate local features of CLIP and the strong global features to effectively harness progressive representational power of models, which enhances the representational capacity of the query intent. Furthermore a dual-combiner architecture is designed with a progressive fusion strategy to fuse multi-granularity complementary and consistent information of reference images and modifying text via the multi-attention module for accurately capturing the query intent. Specifically, for the global combiner, the self-attention is utilized to strengthen relationships among different semantic components within the global feature for improving overall semantic consistency. The cross-attention is designed to establish high-level interactions between the modifying text and the entire image for facilitating holistic understanding and context integration. The channel-attention is applied to adaptively emphasize vital feature channels to the query while suppressing irrelevant channels for enhancing the discriminative power of global features. For the local combiner, we design the self-attention to capture and emphasize critical modification cues within the modifying text. The cross-attention establishes fine-grained and dynamic associations between text words and corresponding local image regions to capture the cross-modal consistency. The channel-attention is designed to highlight visual and text attributes relevant to each other.

Overall, the main contributions of this paper can be summarized as follows:

- We propose a novel Dual-Combiner with Multi-attention (DCMA) network, which employs a progressive fusion strategy that integrates self-attention, cross-attention, and channel-attention mechanisms to better capture the query intent of users.

- The global combiner combines complementary intra- and inter-modal information to capture the contextual semantics.

- The local combiner preserves fine-grained details of the reference image while performing precise local modifications.

- Extensive experiments on standard CIR benchmarks (FashionIQ Wu et al. (2021) and Shoes Guo et al. (2018)) validate the superiority of DCMA, demonstrating substantial performance gains over state-of-the-art methods.

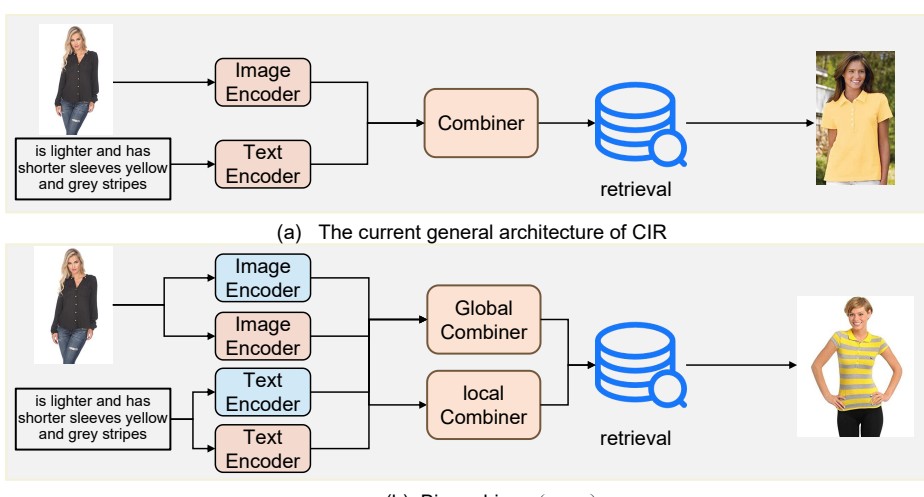

(a) The current general architecture of CIR

(b) Bi-combiner（ours）

Figure 1: Comparison of DCMA and the current general architecture of CIR

## 2 RELATED WORK

### 2.1 COMPOSED IMAGE RETRIEVAL

Composed image retrieval, as a multi-modal retrieval technology, holds significant theoretical importance and immense commercial potential in practical applications such as e-commerce and social media. It substantially improves the accuracy and flexibility of multimodal retrieval by integrating image and text information to meet users' more complex retrieval intentions Hou et al. (2024); Liu et al. (2023); Han et al. (2023b); Xu et al. (2024); Saito et al. (2023); Zhang et al. (2024) . The concept was first introduced by Vo et al., who employed a gated residual module to achieve selective combination of multi-modal queries. This approach improved image retrieval performance by fusing features from both images and text, thereby laying the foundation for subsequent research. Wen et al. designed CLVC-Net to fuse feature representations by combining local and global combination modules. In addition, Goenka et al. (2022) innovatively developed the FashionVLP architecture through visual language transformers to efficiently integrate multi-level fashion-related information. The proliferation of deep learning architectures has driven widespread exploration attention mechanism-based enhancements. For instance, Chen & Bazzani (2020) utilized attention mechanisms to merge hierarchical image features with textual features. These methods effectively focus on key information within both images and text, thereby better addressing the correlation issues between multi-modal data. Ashish (2017) augmented the TIGR by addressing feature divergences between source-target pairs and target images and integrating reference image features with modification textual features at multiple granular levels. Similarly, Wen et al. (2021) fused reference images and modification text at a fine-grained feature hierarchy, a meticulous modeling approach that aids the system in better understanding user requirements.Subsequently, Lee et al. (2021) introduced content modulators and style modulators to update reference images locally and globally based on textual information. This method enables effective adjustments to images at different levels, maximizing inter-modal congruence with hybrid fusion strategies and thereby improving image retrieval performance. Chen et al. (2020) proposed the VAL framework, which embeds composite transformers within convolutional neural networks to selectively retain and modify visual content based on textual feedback. Additionally, the framework optimizes the model through hierarchical matching objectives to further enhance image retrieval effectiveness. Despite the significant advancements these methods have achieved in the field of image retrieval, several limitations remain. Specifically, the reference image and the modifying text are often not directly related but rather exhibit subtle latent correlations. Existing models struggle to fully capture these indirect relationships, leading to insufficient modeling of cross-modal interactions. Moreover, the capability of existing approaches to fuse multi-granular semantic information remains suboptimal, frequently leading to limited performance.

### 2.2 VISION-LANGUAGE MODELS

Models such as CLIP Radford et al. (2021), BLIP Li et al. (2022) , and SigLIP Zhai et al. (2023) have been pre-trained on large-scale web-curated datasets including LAION-400M/5B Schuhmann et al. (2022) and DataComp Li et al. (2024), learning to align visual and textual representations into a unified semantic space. These models serve as foundational backbones for a variety of applications, spanning generative tasks Rombach et al. (2022) Ramesh et al. (2022), open-vocabulary recognition Xu et al. (2023) Wang et al. (2025a), and cross-modal retrieval Wang et al. (2025b). In particular, CLIP leverages contrastive learning on large-scale image-text pairs to achieve robust cross-modal representation capabilities, laying a foundation for a wide range of vision-language tasks,while SigLIP enhances training efficiency and zero-shot performance through a sigmoid-based loss, reducing dependency on global softmax normalization. Although these VLMs have been adapted for Composed Image Retrieval via auxiliary networks Baldrati et al. (2022b) Liu et al. (2024) or task-specific fine-tuning Kim et al. (2021), our approach demonstrates that without introducing additional parameters or fine-tuning, the design of integration scheme combining VLMs can achieve strong CIR performance.

## 3 METHOD

This section first introduces the problem definition for Composed Image Retrieval and presents the overall architecture of the proposed DCMA framework in Section 3.1. Subsequently, Section 3.2 details our multi-layer attention network and the adaptive feature fusion strategy. Finally, Section 3.3 describes the batch-based contrastive loss function employed for training the model.

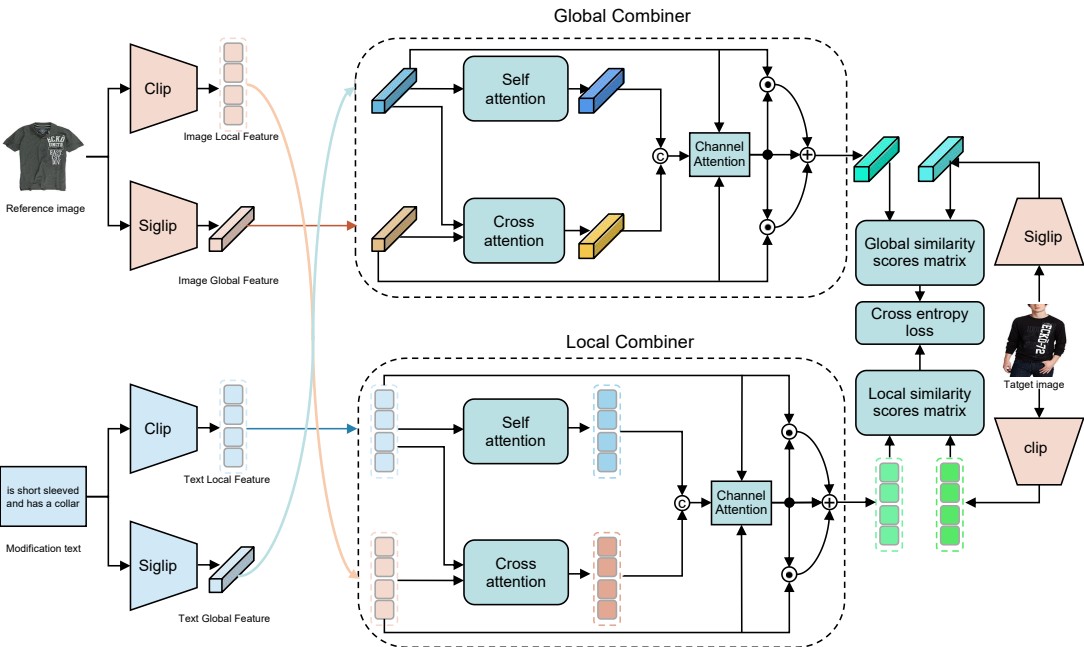

Figure 2: The framework diagram of the Bi-combier method

### 3.1 FRAMEWORK OVERVIEW & PROBLEM DEFINITION

The overall architecture of our proposed framework is illustrated in Figure 2. The Composed Image Retrieval paradigm achieves precise target image identification by fusing multimodal inputs comprising visual exemplars and textual modifications. Formally, let $T$, $I_T$ and $I_R$ denote the modifying text, candidate target image, and reference image, respectively. The composite query structure is defined as : $Q = \langle I_R, T \rangle$.

In this work, we leverage two distinct Vision-Language Pre-trained models for feature extraction. Intermediate layers of the CLIP model are utilized to extract rich local features. Features extracted by CLIP for the reference image, modifying text, and target image are denoted as: $T^C$, $I_T^C$ and $I_R^C$ respectively. For global features, SigLIP is employed to obtain highly discriminative holistic representations, with corresponding features denoted as: $T^S$, $I_T^S$ and $I_R^S$ respectively. By synergistically combining CLIP's detailed local features with SigLIP's highly discriminative global representations, our approach effectively captures both fine-grained patterns and high-level semantic context. Following the acquisition of multi-granularity semantic features, a multi-attention network is designed to precisely fuse the reference image and modifying text. Specifically, the self-attention module is designed to capture the key information in the modifying text, the cross-attention model is employed to capture both the global and local cross-modal semantic consistency, the channel attention module focuses on strengthening vital channels for enhancing the discriminative power of fused features. Finally, the refined features are integrated through an adaptive weighted fusion strategy to generate the final composite representation, which acts as the query embedding for retrieval.

## 3.2 MULTI-LAYER ATTENTION NETWORK AND THE ADAPTIVE FEATURE FUSION STRATEGY

The proposed multimodal architecture employs two dedicated encoder pairs: $\phi_I^C$ and $\phi_T^C$ for CLIP-based feature extraction, and $\phi_I^S$ and $\phi_T^S$ for SigLIP processing. To facilitate cross-modal interaction, a multi-head cross-attention mechanism is implemented, wherein features from the modifying text serve as Query (Q) and features from the reference image function serve as both Key (K) and Value (V). This configuration enables dynamic visual–textual alignment by allowing textual tokens to attend to semantically relevant image regions, as formalized in Equations (1) and (2):

$$MultiHeadCrossAttn(Q, K, V) = Concat(\text{head}_1^{\text{cross}}, \ldots, \text{head}_8^{\text{cross}})\boldsymbol{W}^O \tag{1}$$

$$\text{head}_i^{\text{cross}} = \text{softmax}\left(\frac{\boldsymbol{Q}_i \boldsymbol{K}_i^\top}{\sqrt{d_k}}\right) \boldsymbol{V}_i \tag{2}$$

where the number of attention heads is set to 4, and $\boldsymbol{W}^O$ denotes the output projection weight matrix. Subsequently, a multi-head self-attention mechanism is applied to the modifying text to identify and emphasize crucial elements for image modification, which enables dynamic weighting of critical tokens (e.g., "remove" or "keep") as formalized in Equations (3) and (4):

$$MultiHeadSelfAttn(X) = Concat(\text{head}_1^{\text{self}}, \ldots, \text{head}_h^{\text{self}})\boldsymbol{W}^O \tag{3}$$

$$\text{head}_i^{\text{self}} = \text{softmax}\left(\frac{\boldsymbol{X}_i \boldsymbol{X}_i^\top}{\sqrt{d_k}}\right) \boldsymbol{X}_i \tag{4}$$

where $X \in \mathbb{R}^{n \times d_{\text{model}}}$ denotes the input feature matrix of the modifying text. Subsequently, layer normalization is applied to obtain the normalized text and image features:

$$\hat{\boldsymbol{T}} = \text{LayerNorm}\left(MultiHeadSelfAttn(X)\right) \tag{5}$$

$$\hat{\boldsymbol{I}}_r = \text{LayerNorm}\left(MultiHeadCrossAttn(Q, K, V)\right) \tag{6}$$

$$\boldsymbol{F}_{\text{combC}} = \text{Concat}\left(\hat{\boldsymbol{T}}, \hat{\boldsymbol{I}}_r\right) \tag{7}$$

A channel attention mechanism is implemented to dynamically learn and emphasize the importance of each channel within the input feature maps, as formalized below:

$$\begin{aligned} A_{\text{ref}} &= \text{Sigmoid}\left(\text{FC}_{\text{ref}}\left(\boldsymbol{F}_{\text{combC}}\right)\right) \\ A_{\text{tex}} &= \text{Sigmoid}\left(\text{FC}_{\text{tex}}\left(\boldsymbol{F}_{\text{combC}}\right)\right) \end{aligned} \tag{8}$$

where, FC(.) refers to the stacked fully connected architecture,which is equipped with nonlinear activation and regularization techniques such as ReLU and Dropout. Then the weighted features are then fused as:

$$\boldsymbol{F}_{\text{combA}} = A_{\text{ref}} \odot f_{\text{ref}} + A_{\text{tex}} \odot f_{\text{txt}} \tag{9}$$

where $\odot$ denotes channel-wise multiplication. $f_{\text{ref}}$ and $f_{\text{txt}}$ are features obtained by $\phi_I$ and $\phi_T$. To more effectively capture the implicit associations between the reference image and the modified text, the image-text features are reintroduced, with the scaling factor dynamically adjusted through a learning process. These features are then adaptively combined with the key features via a weighted fusion mechanism, resulting in the final composite representation as shown in Equation (10):

$$\boldsymbol{F}_{\text{comb}} = L(\boldsymbol{F}_{\text{combA}}) + \theta \, f_{ref} + (1 - \theta) f_{txt} \tag{10}$$

where, $L(.)$ represents the dimensional transformation performed by the linear layer, and $\theta$ denotes the dynamically learned weight parameters.

The Multi-attention network precisely aligns the reference image and modifying text while dynamically emphasizing pivotal textual cues, and the adaptive weighted fusion strategy integrates these refined features into a discriminative query embedding, ultimately achieving fine-grained cross-modal interaction and consequently improving retrieval accuracy.

## 3.3 BATCH-BASED CROSS-ENTROPY LOSS

Consistent with prior methodologies, the established Batch-Based Classification loss is employed. Given a batch size B, for the $i$-th query pair embedding $<I_{Ri}, T_i>$ and its corresponding positive target $I_{Ti}$, the forward query loss $L_B$ is computed by:

$$L_{B-CE} = -\frac{1}{B} \sum_{i=1}^{B} \log \left[ \frac{\exp\left[\lambda \cdot \kappa\left(f(I_{Ri}, T_i), I_{Ti}\right)\right]}{\sum_{j=1}^{B} \exp\left[\lambda \cdot \kappa\left(f(I_{Ri}, T_i), I_{Tj}\right)\right]} \right] \tag{11}$$

where $f(,)$ signifies the multimodal feature fusion function, $\lambda$ denotes the temperature hyperparameter that scales the similarity scores before softmax normalization, controlling the sharpness of the probability distribution over positive and negative samples. $\kappa(,)$ represents the cosine similarity kernel, which computes the normalized dot product between two input vectors, measuring their directional alignment in embedding space. Prior to computing the contrastive loss, we implement an adaptive fusion mechanism that integrates global and local similarity representations.The process is formalized as follows:

$$\mathbf{S}_g = \left[\kappa_g\left(f(I_{R_i}, T_i), I_{T_j}\right)\right]_{i,j=1}^{B} \in \mathbb{R}^{B \times B} \tag{12}$$

$$\mathbf{S}_l = \left[\kappa_l\left(f(I_{R_i}, T_i), I_{T_j}\right)\right]_{i,j=1}^{B} \in \mathbb{R}^{B \times B} \tag{13}$$

where, $\mathbf{S}_g$ and $\mathbf{S}_l$ represent the global similarity scores matrix and the local similarity scores matrix respectively. Then integrate them with dynamic weights:

$$\left[w_g^{(i)}, w_l^{(i)}\right] = FC\left(\left[\frac{1}{B} \sum_{j=1}^{B} \mathbf{S}_{g_{ij}}, \frac{1}{B} \sum_{j=1}^{B} \mathbf{S}_{l_{ij}}\right]\right) \tag{14}$$

$$\mathbf{S}_{\text{final}} = \underbrace{\begin{bmatrix} w_g^{(1)} & & \\ & \ddots & \\ & & w_g^{(B)} \end{bmatrix}}_{\mathbf{D}_g} \odot \mathbf{S}_g + \underbrace{\begin{bmatrix} w_l^{(1)} & & \\ & \ddots & \\ & & w_l^{(B)} \end{bmatrix}}_{\mathbf{D}_l} \odot \mathbf{S}_l \tag{15}$$

where, $w_g^{(i)}$ and $w_l^{(i)}$ represent the global and local fusion weights for the $i$-th query, $FC(,)$ denotes a fully-connected network architecture for dynamic weight learning, $\mathbf{D}_g$ and $\mathbf{D}_l$ are diagonal matrices constructed from $w_g^{(i)}$ and $w_l^{(i)}$, $\mathbf{S}_{\text{final}}$ represents the final fused similarity score matrix.

## 4 EXPERIMENTS

This section begins by elaborating on the experimental setup in Section 4.1, followed by a comprehensive performance evaluation against state-of-the-art methods in Section 4.2. Furthermore, Section 4.3 presents ablation studies analyzing the efficacy of our proposed method and the contribution of its individual components. Additionally, Section 4.4 provides qualitative visualizations and discussions to offer deeper insights into the model's behavior.

### 4.1 IMPLEMENTATION DETAIL

Aligned with common practice in the field, our evaluation is conducted on two specialized fashion-domain benchmarks: FashionIQ Wu et al. (2021) and Shoes Guo et al. (2018), allowing for a focused assessment of domain-specific performance. The FashionIQ dataset contains three categories (skirts, shirts and tops), with approximately 46,000 training instances and 15,000 test samples. Based on the acquisition of each triplet, two modification texts were manually annotated, resulting in 18K triplets for training and 6K triplets for testing. The shoe dataset contains 10,751 pairs of shoe images that are characterized by fine-grained distinctions in their attributes. These datasets are challenging as the modifications often involve complex attributes (e.g., color, pattern, style) and relational concepts. Following established practices Delmas et al. (2022) , we use Recall@K (R@K) as our primary evaluation metric. R@K measures the proportion of queries for which the target item appears within

the top-K retrieved results. To provide a comprehensive assessment of retrieval performance on the Fashion-IQ and Shoes datasets, we also report Rmean, defined as the mean across all R@K values. For FashionIQ, Rmean is computed as (R@10 + R@50)/2, while for Shoes, it is the average of R@1, R@10, and R@50.

DCMA is built upon pre-trained CLIP Radford et al. (2021) (ViT-B/32 version) and SigLIP Zhai et al. (2023). The model is optimized using the AdamW optimizer with an initial learning rate of 1e-4 and a batch size of 512. The learnable temperature parameter logit scale is initialized to 100. Empirically, we maintain a consistent projection dimension D of 2048 across the entire network, with the hidden layer dimension set to 4096. All experiments were implemented on a Linux-based system using Python 3.9.20 and PyTorch 2.5.1, and trained for 20 epochs on a single NVIDIA A800 80GB GPU with CUDA 11.8.

Table 1: The performance Comparison on the FashionIQ Dataset

| | Dress | | Shirt | | Toptee | | Average | | Avg |
|---|---|---|---|---|---|---|---|---|---|
| Methods | R@10 | R@50 | R@10 | R@50 | R@10 | R@50 | R@10 | R@50 | Avg |
| TIRG Vo et al. (2019) | 14.87 | 34.66 | 18.26 | 37.89 | 19.08 | 39.62 | 17.40 | 37.39 | 27.40 |
| VAL Chen et al. (2020) | 22.53 | 44.00 | 22.38 | 44.15 | 27.53 | 51.68 | 24.15 | 46.61 | 35.40 |
| MAAF Dodds et al. (2020) | 23.8 | 48.6 | 21.3 | 44.2 | 27.9 | 53.6 | 24.3 | 48.8 | 36.6 |
| CIRPLANT Liu et al. (2021) | 14.38 | 34.66 | 13.64 | 33.56 | 16.44 | 38.34 | 14.82 | 35.52 | 25.17 |
| CoSMo Lee et al. (2021) | 25.64 | 50.30 | 24.90 | 49.18 | 29.21 | 57.46 | 26.58 | 52.31 | 39.45 |
| FashionVLP Goenka et al. (2022) | 32.42 | 60.29 | 31.89 | 58.44 | 38.51 | 68.79 | 34.27 | 62.51 | 48.39 |
| ARTEMIS Delmas et al. (2022) | 25.68 | 51.25 | 28.59 | 55.06 | 21.57 | 44.13 | 25.25 | 50.08 | 37.67 |
| CRN Yang et al. (2023) | 32.67 | 59.30 | 30.27 | 56.97 | 37.74 | 65.94 | 33.56 | 60.74 | 47.15 |
| MGUR Chen et al. (2024) | 32.61 | 61.34 | 33.23 | 62.55 | 41.40 | 72.51 | 35.75 | 65.47 | 50.61 |
| SyncMask Song et al. (2024) | 33.76 | 61.23 | 35.82 | 62.12 | 44.82 | 72.06 | 38.13 | 65.14 | 51.64 |
| LF-CLIP Baldrati et al. (2022a) | 31.63 | 31.63 | 36.36 | 58.00 | 38.19 | 62.42 | 35.39 | 59.03 | 47.21 |
| CLIP4CIR Baldrati et al. (2022b) | 33.81 | 59.40 | 39.99 | 60.45 | 41.41 | 65.37 | 38.32 | 61.74 | 50.03 |
| Prog. Lrn. Zhao et al. (2022) | 38.18 | 64.50 | 48.63 | 71.54 | 52.32 | 76.90 | 46.38 | 70.98 | 58.68 |
| FAME-ViL Han et al. (2023a) | 42.19 | 67.38 | 47.64 | 68.79 | 50.69 | 73.07 | 46.84 | 69.75 | 58.30 |
| FashionSAP Han et al. (2023b) | 33.71 | 60.43 | 41.91 | 70.93 | 33.17 | 61.33 | 36.26 | 64.23 | 50.25 |
| BLIP4CIR Liu et al. (2024) | 40.65 | 66.34 | 40.38 | 64.13 | 46.86 | 69.91 | 42.63 | 66.79 | 54.71 |
| BLIP4CIR+Bi Liu et al. (2024) | 42.09 | 67.33 | 41.76 | 64.28 | 46.61 | 70.32 | 43.49 | 67.31 | 55.40 |
| SSN Yang et al. (2024) | 34.36 | 60.78 | 38.13 | 61.83 | 44.26 | 69.05 | 38.92 | 63.89 | 51.41 |
| PAIR Fu et al. (2025) | 46.78 | 70.93 | 52.60 | 73.80 | 58.91 | 78.81 | 52.76 | 74.51 | 63.64 |
| CCIN Tian et al. (2025) | 49.38 | 72.09 | 53.83 | 74.14 | 58.13 | 78.58 | 53.78 | 74.94 | 64.36 |
| **DCMA(Ours)** | **52.62** | **75.11** | **62.51** | **79.24** | **63.59** | **83.02** | **59.57** | **79.12** | **69.35** |

## 4.2 Performance Comparison

We evaluated various methods on the test and validation sets of the Fashion and Shoes datasets. The reported results represent the average of three independent training runs with different random seeds. As shown in Tables 1 and 2, which compare traditional model-based baselines with VLM-based baselines, our analysis yields three key observations; the best result in each comparison is highlighted in bold. DCMA achieves the best performance across both datasets. Specifically, on the fashion-domain dataset FashionIQ, it improves the Avg by 7.8%, while on the Shoes dataset, it increases the Avg by 3.9%. These results demonstrate its strong generalization capability in domain-specific scenarios such as fashion. Vision-language pre-training (VLP) based models—such as MMF and FAME-ViL consistently and significantly outperform conventional methods, including TIRG and CLVC-Net, across multiple experimental settings. This advantage can be attributed to the powerful representational capacity acquired by VLP models through large-scale cross-modal pre-training, which enables them to more effectively capture fine-grained semantic correspondences between images and text. Consequently, these results underscore the substantial potential and practical value of VLP-based approaches in fine-grained cross-modal understanding tasks. DCMA significantly enhances cross-modal semantic alignment and feature fusion by effectively integrating the

complementary strengths of two distinct Vision-Language Pre-training (VLP) models, coupled with a multi-granularity information interaction mechanism.

Table 2: The performance Comparison on the Shoes Dataset

| | Shoes | | | |
|---|---|---|---|---|
| Methods | R@1 | R@10 | R@50 | Avg |
| TIRG Vo et al. (2019) | 12.60 | 45.45 | 69.39 | 42.48 |
| FashionVLP Goenka et al. (2022) | - | 49.08 | 77.32 | - |
| ARTEMIS Delmas et al. (2022) | 18.72 | 53.11 | 79.31 | 50.38 |
| MGUR Chen et al. (2024) | 18.41 | 53.63 | 79.84 | 50.63 |
| Prog. Lrn. Zhao et al. (2022) | 22.88 | 58.83 | 84.16 | 55.29 |
| PAIR Fu et al. (2025) | 27.48 | 65.08 | 85.86 | 59.47 |
| CCIN Tian et al. (2025) | 25.95 | 65.76 | **86.54** | 59.42 |
| **DCMA(Ours)** | **31.80** | **67.75** | 86.43 | **61.99** |

## 4.3 ABLATION STUDY

To elucidate the contribution of each component in DCMA, we conducted an ablation study by evaluating three model variants: "Bi w/o local combine" (removing the local combine stream), "Bi w/o global combiner" (removing the global combiner stream), and "Bi w/o combiner" (replacing the multi-attention combiner with a simple feature combination strategy). The results, summarized in Table 3, provide clear insights into the role of each component. We observe that removing the SigLIP stream leads to the most significant performance degradation, highlighting its importance in capturing the global context of user's query intent. Similarly, removing the CLIP stream results in a noticeable performance decline, which highlights the critical importance of fine-grained local features in maintaining visual consistency between reference and target images. Furthermore, the retrieval performance decreases when the combiner network is substituted by the simple feature combination strategy, which verifies the effectiveness of the proposed multi-attention fusion mechanism. Collectively, these results affirm that each component plays a crucial and distinct role in the overall performance of DCMA.

Table 3: The ablation study on FashionIQ validation data demonstrates optimal performance metrics.

| | Dress | | Shirt | | Toptee | | Average | | Avg |
|---|---|---|---|---|---|---|---|---|---|
| Methods | R@10 | R@50 | R@10 | R@50 | R@10 | R@50 | R@10 | R@50 | Avg |
| **Bi w/o local combiner** | 51.51 | 74.12 | 62.51 | 78.85 | 62.92 | 80.92 | 58.98 | 77.96 | 68.47 |
| **Bi w/o global combiner** | 42.19 | 67.08 | 42.20 | 64.47 | 47.78 | 71.44 | 44.06 | 67.66 | 55.86 |
| **Bi w/o combiner** | 49.23 | 71.44 | 61.53 | 76.84 | 61.86 | 79.96 | 57.54 | 76.08 | 66.81 |
| **DCMA** | **52.62** | **75.11** | **62.51** | **79.24** | **63.59** | **83.02** | **59.57** | **79.12** | **69.35** |

## 4.4 QUALITATIVE VISUALIZATION

As shown in Figure 3, we compare the top-5 retrieval results across three categories of the FashionIQ dataset. It can be observed that the baseline model fails to retrieve a "darker" shade of blue as required. Furthermore, it incorrectly returns a dress with a black background instead of a white-based one. Finally, it also retrieves a blue top/tee when the desired target is green. These visual examples vividly demonstrate DCMA's superior ability in fine-grained attribute binding and cross-modal alignment, effectively translating textual modifications into precise visual changes while preserving the style of the reference image. In contrast, our model demonstrates enhanced capabilities in cross-modal interaction, captures more fine-grained discriminative features, and retrieves target images that better align with the user's intent.

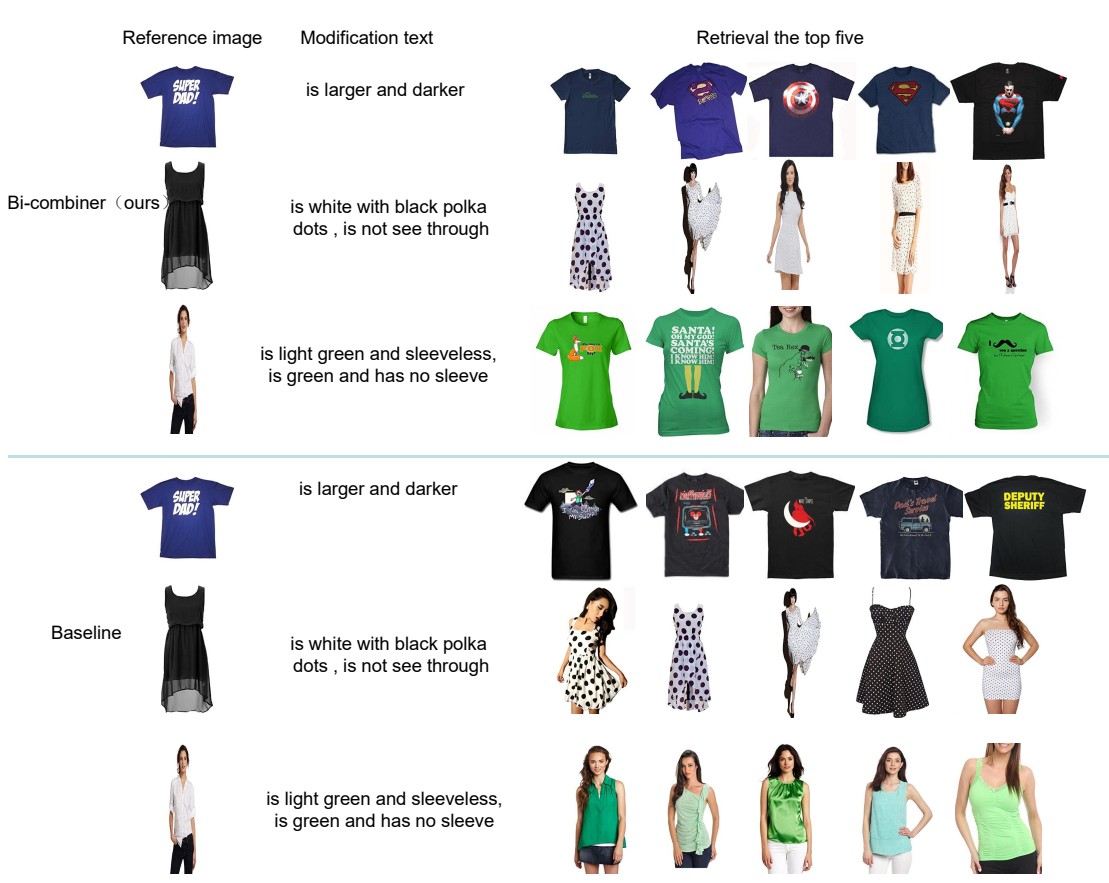

Figure 3: Qualitative image retrieval results on three categories of FashionIQ. We mainly compare the top-5 ranking list of the proposed method with the baseline.

## 5 CONCLUSION

In this work, we proposed DCMA, a novel dual-stream architecture for compositional image retrieval. DCMA integrates self-attention, cross-attention, and channel-attention mechanisms within a multi-attention framework. Its global combiner captures holistic contextual semantics, while the local combiner preserves fine-grained visual consistency. The contribution of DCMA to cross-modal interaction modeling was investigated and evaluated through extensive experiments on benchmark datasets, complemented by qualitative visualizations. We hope that this method will advance the development of compositional image retrieval, and we plan to explore more sophisticated cross-modal fusion strategies in future work.

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
