# OpenReview forum: "Dual-Combiner Network with Multi-Attention for Composed Image Retrieval"
_ICLR.cc/2026/Conference — Submitted to ICLR 2026_

### Official Review · Reviewer_GwVx · 2025-10-25

**Soundness:** 2
**Presentation:** 3
**Contribution:** 1
**Rating:** 2
**Confidence:** 4

**Summary:**

The paper introduces Dual-Combiner with Multi-Attention (DCMA), a network for Composed Image Retrieval (CIR), which aims to retrieve images based on a reference image and a textual modification. The proposed method integrates CLIP and SigLIP for local/global feature representations, combining them through self/cross/cannnel attention modules in two parallel combiners.

**Strengths:**

- The authors describe the method architecture in detail.
- Strong empirical results on two CIR benchmarks.

**Weaknesses:**

## Limited Contribution ##
- The paper’s main contribution is narrow and insufficiently distinguished from prior work. It overlooks [1], which already introduced cross- and channel-attention modules specifically for CIR. The concluding statement of the Related Work section - "Moreover, the capability of existing approaches to fuse multi-granular semantic information remains suboptimal" does not acknowledge this prior work. The authors should clarify how their method conceptually differs from [1], as both aim to enhance bimodal fusion via cross/channel attention, even if the implementations differ.

- The contribution list at the end of the Introduction section is misleadingly structured: the two middle points describe method characteristics rather than genuine contributions, and should be merged with the first item.

- The final claimed contribution “Extensive experiments on standard CIR benchmarks” is inaccurate. 'Shoes' is not a standard CIR benchmark in current literature; more relevant ones are CIRCO, CIRR, and FashionIQ (the latter being included in this paper). I assume the authors know these benchmarks from the papers they cite (in Table 1 for example). This discrepancy is notable given that the proposed method uses general-domain backbones (SigLIP, CLIP) and is not restricted to fashion-specific datasets.

Considering these issues, the overall novelty and contribution of the paper appear limited.

## Related Work ##
Line 135 states: "Chen et al. (2020) proposed the VAL framework, which embeds composite transformers within convolutional neural networks to selectively retain and modify visual content based on textual feedback". This description reads like a generic LLM-generated summary and does not demonstrate understanding of the referenced work. The "based on textual feedback" imply for one of the fundamental use-cases for CIR, so it like saying "this paper address CIR". The authors’ explanation adds no meaningful context.

## Paper Writing ##
- Lines 160–161 contain a contradiction: the authors claim that their approach does not involve training, while Section 3 explicitly describes a training process.
- Line 048: "Despite these advances, CIR remains highly challenging due to the inherent semantic and modal gaps between images and text, which hinder accurate modeling of joint retrieval intent" - this claim lacks supporting evidence or citation, the authors should justify or reference it. Note that Image2Image Retrieval is also challenging, but we can safely say that there is no modality gap between image and image.


## Evaluation ##
- The paper omits a baseline evaluation of the chosen backbones. It is unclear whether performance gains stem from the proposed architecture or simply from using a stronger pre-trained model (SigLIP). Table 1 lists multiple CIR methods based on large vision-language models (e.g., CLIP4CIR, BLIP4CIR) that could potentially surpass the proposed approach if similarly updated to SigLIP (e.g., SigLIP4CIR).
- Relatedly, the method combines two separate backbones (CLIP and SigLIP) but compares against works using only one. This raises a fairness concern - improvements might primarily arise from increased model capacity rather than the dual-combiner design. The paper would be improved with, for example, a control experiment using a single backbone twice for a fair comparison.

### Minor comments ###
- Fix spacing before/after citations/periods (e.g., lines 038, 053, 131).
- The paper inconsistently refers to the same objective as both Batch-Based Classification Loss and Contrastive Loss. The standard term in literature is Contrastive Loss and should be used consistently.

[1] Levy, M., Ben-Ari, R., Darshan, N., & Lischinski, D. (2024). Data Roaming and Quality Assessment for Composed Image Retrieval. Proceedings of the AAAI Conference on Artificial Intelligence, 38(4), 2991-2999.

**Questions:**

- Line 139: “Specifically, the reference image and the modifying text are often not directly related but rather exhibit subtle latent correlations” - not clear. What does it mean? Based on what? Any reference?

---

### Official Review · Reviewer_NepZ · 2025-10-27

**Soundness:** 2
**Presentation:** 2
**Contribution:** 2
**Rating:** 2
**Confidence:** 4

**Summary:**

The proposed DCMA network, with its dual-combiner architecture and multi-attention fusion mechanism, effectively addresses key challenges in the CIR task. The global combiner captures overall contextual semantics, while the local combiner preserves fine-grained visual consistency. The method achieves promising results across multiple benchmarks.

**Strengths:**

1. Leverages the strengths of CLIP (excelling at local details) and SigLIP (excelling at global discrimination) instead of relying on a single model, ensuring the capture of multi-granularity information at the feature extraction level;

2. Designs a multi-attention combiner that progressively fuses self-attention, cross-attention, and channel attention;

3. Introduces no additional trainable parameters (only utilizing pre-trained VLPMs) and requires no fine-tuning of the VLPMs;

**Weaknesses:**

1. The so-called local-global learning has been extensively studied in multi- and cross-modal learning, such as word-patch or sentence-image alignment. This work does not provide new insights in this regard;

2. Before computing the final loss, the authors design an adaptive fusion mechanism. However, it is unclear why dynamic weighting of global and local similarities is necessary, rather than simply summing or concatenating them. In retrieval tasks, a fixed fusion weight or a static weight learned through a network might already suffice;

3. It is unclear whether the performance improvement of DCMA relies on specific attributes of fashion items, such as regular variations in color, texture, or shape. Would the method be equally effective in more complex and semantically rich scenarios, such as natural images, interior design, or abstract concept retrieval?

4. The paper does not provide any key efficiency metrics for the model, such as inference speed (FPS), parameter count, or FLOPs. In real-world retrieval systems, efficiency is as important as accuracy. It remains unclear whether the performance gains of DCMA justify its additional computational overhead compared to simpler approaches using a single VLPM backbone, such as directly using CLIP or SigLIP with simple fusion.

**Questions:**

See the Weaknesses.

---

> ### Comment · Reviewer_NepZ · 2025-11-26
>
> It appears that the authors did not submit a rebuttal. Taking into account the evaluations from the other reviewers, I maintain my original score.

---

### Official Review · Reviewer_3VFf · 2025-10-31

**Soundness:** 3
**Presentation:** 3
**Contribution:** 2
**Rating:** 2
**Confidence:** 4

**Summary:**

In this paper, the authors focused on composed image retrieval. Considering the existing studies overlooked the consistency of unmentioned regions with the target image, the authors proposed a dual-combiner with a multi-attention network for composed image retrieval. Specifically, the designed method comprises a global combiner and a local combiner, where the former captures global contextual information and the latter extracts fine-grained local details. Experimental results prove the effectiveness of the proposed method.

**Strengths:**

1. The proposed method integrates multiple fusion strategies to achieve more comprehensive multimodal fusion.
2. The integration of CLIP and SigLIP enhances the feature extraction capability for both query and target.

**Weaknesses:**

1. The proposed method lacks significant innovation. The integration of local and global modules has been widely adopted in existing literature, as exemplified by CLVC-Net [1] and IUDC [2]. Given this, the method is somewhat incremental.
[1] Comprehensive Linguistic-Visual Composition Network for Image Retrieval. SIGIR 2021
[2] LLM-Enhanced Composed Image Retrieval: An Intent Uncertainty-Aware Linguistic-Visual Dual Channel Matching Model. ACM TOIS 2025
2. All datasets selected for evaluation are confined to the fashion domain, meaning the model’s generalization capability in open-domain scenarios (such as the CIRR dataset [3]) remains unverified.
[3] Image retrieval on real-life images with pre-trained vision-and-language models. ICCV 2021

**Questions:**

As listed above.

---

### Official Review · Reviewer_6FSh · 2025-11-01

**Soundness:** 2
**Presentation:** 3
**Contribution:** 2
**Rating:** 2
**Confidence:** 5

**Summary:**

This paper addresses Composed Image Retrieval (CIR) by proposing a Dual-Combiner with Multi-Attention (DCMA) architecture. The core idea is to fuse multi-granularity features from two pre-trained vision-language models (intermediate CLIP features as local cues and SigLIP global representations) via two parallel combiners: a global combiner (self-, cross-, and channel-attention to model global context) and a local combiner (fine-grained region–token interaction). Experiments on FashionIQ and Shoes show consistent improvements over a set of prior methods, and ablations claim each module contributes to the gain.

**Strengths:**

1.The paper is well-structured, with clear explanations of the methodology, equations, and figures. The problem definition and framework overview are particularly accessible.

2.The integration of CLIP and SigLIP for complementary feature extraction, combined with a dual-combiner architecture and multi-attention fusion. The progressive fusion strategy effectively bridges modal gaps and captures both local and global semantics.

**Weaknesses:**

1. The proposed architecture shows limited methodological novelty. Both the Local Combiner and the Global Combiner share almost identical structures, differing mainly in their input features—one derived from CLIP and the other from SigLIP. Conceptually, the design can be interpreted as performing two rounds of inference using different backbone encoders, followed by a weighted similarity aggregation. As such, the method appears to be an architectural reuse rather than a fundamentally new compositional learning framework.

2. The experiments are restricted to fashion-domain datasets (FashionIQ and Shoes) and do not include evaluations on more general-purpose or open-domain benchmarks such as CIRR [1], which is a standard dataset for assessing compositional image retrieval performance under real-world and diverse scenarios. The absence of such experiments raises concerns about the proposed method’s generalization ability to open-domain or real-world retrieval tasks.

3. The use of dual-model backbones (CLIP and SigLIP) combined with multiple attention modules likely introduces substantial computational overhead. However, the paper does not provide any analysis of inference speed, memory footprint, or computational complexity. Since deployment efficiency is crucial for practical retrieval systems, the lack of this analysis undermines the paper’s claims of applicability in real-world scenarios.

4. The paper lacks sufficient visual analysis to support the quantitative results. For instance, retrieval visualizations are not provided, and Figure 3 omits explicit marking of the ground-truth target image, which weakens the interpretability and persuasiveness of the results.

5. The internal design of the proposed combiners is not empirically analyzed. There is no ablation experiment to examine the contribution of each attention component or fusion module, leaving the rationale for the current structural choices unexplained.

6. In Section 4.3, the authors mention replacing the “multi-attention combiner with a simple feature combination strategy” for comparative analysis, but the paper does not clarify what the “simple feature combination” specifically entails. The lack of detailed description prevents a fair and reproducible comparison.

7. In Table 3, given that both combiners share an identical structure, the difference between “Bi w/o local combiner” and “Bi w/o global combiner” is essentially the use of different backbone encoders. The observed performance gains thus seem more attributable to the backbone change rather than the architectural design itself, suggesting that the reported improvement might stem from stronger encoders instead of the proposed technique.

8. What would be the result if the Local Combiner used the final output features of the CLIP model instead of its intermediate layer features?

[1] Liu, Z., Rodriguez-Opazo, C., Teney, D., Gould, S., 2021b. Image retrieval on real-life images with pre-trained vision-and-language
models, in: Proceedings of the IEEE/CVF International Conference on Computer Vision, pp. 2125–2134.

**Questions:**

Please the part of "Weakness"

---

### Meta-Review · Area_Chair_2Ba8 · 2025-12-26

**Summary:**

6FSh: (1) novelty is limited. (2) Experimented are limited to fashion-domain datasets. (3) Substantial computational overhead. (4) Lack of sufficient visual analysis. (5) Lack of empirical analysis/ablation. (6) Lack of detailed description for "simple feature combination". (7) The performance improvement may come from stronger encoders (not the proposed technique.

3VFf: (1) lack of innovation. (2) Limited datasets used in experiment.

NepZ: (1) Limited novelty. (2) Whether the dynamic fusion mechanism is necessary. (3) Whether the performance improvement can be generalized to more complex scenarios (beyond fashion items). (4) lack of certain key efficiency metrics

GwVx: (1) limited contribution. (2) various issues in writing. (3) insufficient evaluation

All reviewers recommended "2:reject". There is no rebuttal from the authors. So there is no reason to overrule reviewers' recommendation

**Reviewer Concerns:**

The authors did not provide rebuttal. So the reviewers' concerns are still outstanding.

**Reviewer Scores:**

Since no rebuttals are provided, the scores are unlikely to change.

---

### Decision · Program_Chairs · 2026-01-26

Reject